# Enoxaparin Posology According to Prothrombotic Status and Bleeding Risk in Hospitalized Patients with SARS-CoV-2 Pneumonia

**DOI:** 10.3390/jcm12030928

**Published:** 2023-01-25

**Authors:** Juan Mora-Delgado, Cristina Lojo-Cruz, Patricia Rubio Marín, Eva María Menor Campos, Alfredo Michán-Doña

**Affiliations:** 1Internal Medicine and Palliative Care Clinical Management Unit, Hospital Universitario de Jerez de la Frontera, Jerez de la Frontera, 11407 Cádiz, Spain; 2Departamento de Medicina, Biomedical Research and Innovation Institute of Cadiz (INiBICA), Avenida Ana de Viya, 21, 11009 Cádiz, Spain

**Keywords:** low molecular weight heparin, enoxaparin, COVID-19, SARS-CoV-2 pneumonia, thromboembolic disease, thrombotic risk, hemorrhagic risk

## Abstract

Some patients with COVID-19 have complex hypercoagulable abnormalities that are related to mortality. The optimal dosage of low molecular weight heparin in hospitalized patients with SARS-CoV-2 pneumonia is still not clear. Our objective is to evaluate the effects of adapting the dosage of low molecular weight heparin to thrombotic and bleeding risk scales in this setting. We performed a cohort, retrospective, observational, and analytical study at the Hospital Universitario of Jerez de la Frontera, with patients admitted with SARS-CoV-2 pneumonia from 1 October 2020 to 31 January 2021. They were classified according to whether they received prophylactic, intermediate, or therapeutic doses of enoxaparin. The primary endpoint was intrahospital mortality. Secondary endpoints were the need for invasive ventilation, thromboembolic events, bleeding, and the usefulness of thrombotic and bleeding scales. After binary logistic regression analysis, considering confounding variables, it was found that the use of enoxaparin at therapeutic doses was associated with lower mortality during admission compared to prophylactic and intermediate doses (RR 0.173; 95% CI, 0.038–0.8; *p* = 0.025). IMPROVE bleeding risk score correlated with a higher risk of minor bleeding (RR 1.263; 95% CI, 1.105–1.573; *p* = 0.037). In adult hospitalized patients with SARS-CoV-2 pneumonia presenting elevated D-dimer and severe proinflammatory state, therapeutic doses of enoxaparin can be considered, especially if bleeding risk is low according to the IMPROVE bleeding risk score.

## 1. Introduction

Patients with COVID-19 present various clinical manifestations and complications. Although respiratory symptoms are an important characteristic problem of COVID-19 [1], some patients have complex hypercoagulable abnormalities that are closely related to mortality [2,3]. The hypercoagulable state can be induced due to variations in prothrombotic factors, such as elevated fibrinogen, D-dimer, and factor VIII levels. These coagulation abnormalities suggest a hypercoagulable state different from disseminated intravascular coagulopathy and vaccine-induced immune thrombotic thrombocytopenia.

Heparin may have beneficial mechanisms of action in COVID-19 patients, such as direct antiviral effectiveness against SARS-CoV-2. SARS-CoV-2 fusion and entrance into the host cell [4,5], as in the Severe Acute Respiratory Syndrome Coronavirus (SARS-CoV) and Middle Eastern Respiratory Syndrome Coronavirus (MERS-CoV) viruses, is assumed to be mediated by the binding of viral spike proteins to the host via ACE2 receptors [6]. Cell surface heparan sulfate, a kind of glycosaminoglycan found in heparin, has been shown in vitro to be necessary for human coronavirus NL63 and SARS-CoV entry and infectivity. As a co-receptor for adhesion molecules, heparan sulfate is thought to interact with the spike protein, which might be the initial step toward promoting SARS-CoV-ACE2 contact [7].

The usefulness of heparin as an anticoagulant in COVID-19 was first addressed in a retrospective report of 449 COVID-19 patients from Wuhan, China, where prophylaxis in medical patients is relatively uncommon due to the low incidence of VTE. In this cohort, 350 patients received no heparin therapy (neither low-dose prophylaxis nor high-dose therapy), while 99 received low-dose prophylactic heparin therapy. Patients with elevated D-dimer levels (six times the upper limit of normal) or sepsis-induced coagulopathy who received prophylactic heparin had approximately 20% lower mortality than patients who did not [8]. Intravenous tissue plasminogen activator, a potent thrombolytic, is also reported to transiently improve oxygenation in COVID-19-related acute respiratory distress syndrome, supporting the clinical relevance of thrombosis in severe disease [9].

In a nationwide cohort study of 4297 patients with COVID-19 in the United States that compared no anticoagulant versus administration of heparin as a prophylactic anticoagulant within the first 24 h, reduced mortality rates were observed [10]. A multicenter retrospective observational study in patients with COVID-19 in Italy showed that heparin use was independently associated with reduced mortality rates, particularly in patients over 59 years of age [11].

These clinical data provided strong real-world evidence to support guidelines recommending the use of, at least, prophylactic anticoagulants as first-line treatment for patients with COVID-19 at hospital admission.

Several studies have compared the effects of therapeutic and prophylactic or intermediate-dose anticoagulants. Some reported that low-dose anticoagulants do not significantly improve clinical outcomes such as the need to use invasive or non-invasive mechanical ventilation, ICU admission, and death compared to intermediate- or therapeutic-dose anticoagulants [12,13,14,15]. On the other hand, other studies have reported that therapeutic doses of LMWH could reduce thromboembolic events and deaths compared to prophylactic treatment or treatment with intermediate doses of heparin in high-risk COVID-19 patients with high D-dimer levels [16,17,18,19].

Currently, most guidelines do not recommend the routine use of therapeutic doses of anticoagulants [20]. Additionally, other alternatives are being evaluated as anticoagulants that may modify the course of COVID-19. A pilot multicenter randomized controlled trial aimed to determine if bivalirudin was an effective alternative to enoxaparin in reducing the time spent under invasive mechanical ventilation in COVID-19 patients at risk of thromboembolic complications. Further research is needed to establish its effectiveness in this population [21].

The main objective of this study is to determine the effectiveness of adjusting LMWH dosage based on prothrombotic conditions to reduce mortality and the need for invasive mechanical ventilation in patients hospitalized with SARS-CoV-2 pneumonia. Furthermore, we aim to evaluate the impact of this dosage adjustment on the incidence of thromboembolic and hemorrhagic events. In addition, the study aims to assess the reliability of different thrombotic and hemorrhagic risk scales in the context of COVID-19. By combining clinical, laboratory, and risk scale data, the study aims to provide solid evidence to aid in clinical decision making.

## 2. Materials and Methods

### 2.1. Study Design

A cohort, retrospective, observational, and analytical study was conducted at the Hospital Universitario of Jerez de la Frontera with patients admitted for SARS-CoV-2 pneumonia (confirmed by a PCR or antigen test) between 1 October 2020 and 31 January 2021 (second and third waves in Spain). Patients were classified according to whether they received prophylactic doses (40 mg/24 h) versus intermediate (1 mg/kg/24 h) and therapeutic doses (1 mg/kg/12 h) of enoxaparin according to our hospital protocol (Figure 1).

### 2.2. Patients

Inclusion criteria were patients older than or equal to 18 years of age with a positive PCR or antigen test for SARS-CoV-2 and pneumonia evidenced by a chest X-ray or CT that were admitted to the Hospital Universitario de Jerez de la Frontera.

Exclusion criteria were patients with chronic outpatient therapeutic anticoagulation, an absence of data that would prevent the adequate collection of study variables, patients that were clinically judged at the discharge of infection with a high suspicion of COVID-19 without microbiological confirmation from a PCR or antigen test, or patients with a hereditary history of procoagulant diathesis.

### 2.3. Selection of Thrombotic and Hemorrhagic Risk Scales

We carefully selected a number of thrombotic and bleeding risk scales that we deemed useful for our study. These scales were chosen because we believe they effectively reflect and take into account factors related to coagulopathy caused by COVID-19 [23]. The scales that we have chosen and summarized below, along with the relevant evidence regarding their use in relation to COVID-19, are the Caprini risk score, Padua prediction score, PRETEMED score, and IMPROVE bleeding risk score.

The Caprini risk score is a risk assessment tool for the occurrence of venous thromboembolism among surgical patients. The Caprini risk score includes 20 variables and it is derived from a prospective study of 538 general surgery patients [24]. A score equal to or greater than 3 implies a high risk of VTE. The Caprini score has already been endorsed by the American Venous Forum for VTE risk assessment at admission and discharge. It has been validated in over 250,000 patients in more than 100 clinical trials worldwide [25]. Some studies found that, despite the administration of LMWH, there was a significant correlation between the Caprini score and the risk of VTE in patients with COVID-19 [26].

Padua prediction score (PPS) is a risk assessment model created by modifying the initial Kucher model, which is used to calculate the risks of VTE in hospitalized medical patients without medical prophylaxis. The PPS, developed in 2010, is a risk prediction model that identifies patients’ propensity to VTE. It is made up of 11 risk variables, each of which is assigned a score from 1 to 3; the authors assigned low risk to those with a score lower than 4 points and high risk to those with a score of 4 or more [27]. PPS is an independent risk factor for the in-hospital mortality of COVID-19 patients. Moreover, prophylactic anticoagulation in patients with higher PPS scores showed a mild advantage in terms of mortality without statistical significance. This demonstrated that PPS could be a promising tool for evaluating early prognosis in COVID-19 patients [28].

The PRETEMED score was established in 2003 by the Andalusian Society of Internal Medicine (SADEMI) and the Andalusian Society of Family and Community Medicine (SAMFYC) and was updated in 2007 for the evaluation of antithrombotic prophylaxis in patients with acute or chronic medical conditions. This scale makes recommendations based on the combination of risk variables by weighing thrombotic risk. It assigns a number between 1 and 3 and distinguishes between precipitating processes, medication, associated processes, and those connected with the development of VTE. When the result is between 1 and 3, general measures or mechanical prophylaxis are advised; when the result is 4, pharmaceutical prophylaxis with LMWH is advised [29]. We have not found studies that evaluate the applicability of this scale to patients with COVID-19.

The IMPROVE bleeding risk score was defined through a multicenter study designed to evaluate prophylactic practices for venous thromboembolic disease and clinical outcomes in hospitalized patients between July 2002 and September 2006. Data from the IMPROVE registry were utilized while developing this scale to examine the rate of in-hospital bleeding and identify admission risk variables related to bleeding risk in severely ill medical patients. This score was constructed using 13 risk indicators to determine the probability of bleeding. A value higher than 7 indicated a high risk of bleeding (4%) [30]. So far, this score has shown moderate predictive ability and an acceptable negative predictive value for bleeding [31,32].

### 2.4. Variables

Independent qualitative variables were gender, smoking habit, cardiovascular risk factors (high blood pressure, diabetes mellitus, dyslipidemia, and obesity), respiratory pathology (bronchial asthma, chronic obstructive pulmonary disease, and sleep apnea/hypopnea syndrome), chronic kidney disease, previous comorbidities treatment, radiological pattern as defined by the modified RALE score [33] on admission, and dosages of enoxaparin (prophylactic, intermediate, or therapeutic dose) with concomitant treatments received (dexamethasone and methylprednisolone, tocilizumab, anakinra, ruxolitinib, remdesivir) during hospitalization. Independent quantitative variables were age, relationship between peripheral O_2_ saturation and FiO_2_, lowest value of lymphocytes and prothrombin time (%), together with the maximum value of D-dimer, thromboplastin time (ratio), lactate dehydrogenase, C-reactive protein, procalcitonin, and ferritin, CALL SCORE [22], and thrombotic and hemorrhagic risk scales (PRETEMED, Padua, and Caprini scales and IMPROVE Bleeding Risk Assessment Score). Dependent variables included deep vein thrombosis (evidenced by doppler ultrasound), pulmonary thromboembolism (evidenced by angioCT), hemorrhagic events (stratification of severity according to the ISTH scale in major or minor events [34]), primary outcome (discharge or death), and secondary outcomes (admission to the ICU, need for invasive mechanical ventilation and use of high-flow nasal cannula oxygen) during hospitalization.

### 2.5. Statistical Analysis

Initially, a bivariate analysis was carried out prior to the multivariate analysis so that the independent variables could be compared with respect to thromboembolic events, hemorrhagic events, and outcomes. Quantitative variables with normal and abnormal distribution were compared using the Student’s t-test or Mann–Whitney U-test, respectively. Categorical variables were compared using the chi-square test. The results were expressed as median plus interquartile range or percentage, as appropriate. Those variables that showed statistical significance in the bivariate analysis were included in a logistic regression analysis to determine which were confounding variables and which were effect modifiers. At the end of the analysis, the adjusted risks could be calculated for each independent variable after the existence of interaction and confusion had been assessed. Specifically, for intergroup statistical analysis (Table 1), chi-square test were used for categorical variables and Student’s t-test for continuous variables. A binary logistic regression analysis was conducted to examine the relationship between independent variables and in-hospital mortality, which was used as the dependent variable (Table 2). In order to evaluate the prothrombotic and hemorrhagic risk scales in relation to different dosages of LMWH, a multinomial logistic regression analysis was conducted (Table 3). A value of *p* < 0.05 was considered statistically significant. Data were analyzed with SPSS 22.0 for Windows (SPSS Inc., Chicago, IL, USA).

### 2.6. Ethics

Approval was granted by the Cádiz Research Ethics Committee on 9 June 2022.

An exemption from informed consent was approved since this was an observational study with significant social value and its realization would not be feasible or viable without such renunciation, given the high percentage of deceased patients and the long study period.

## 3. Results

From 1 October 2020 to 31 January 2021, 676 patients were coded on discharge at the Hospital Universitario de Jerez de la Frontera as “COVID-19”, of which 100 patients were admitted to other hospital units due to other pathologies and/or did not present pneumonia, 78 patients presented atrial fibrillation/flutter, 7 patients were previously anticoagulated due to venous thromboembolic disease, 9 patients were underage, 1 patient was anticoagulated due to a biological prosthetic valve, and 20 patients did not have the minimum data necessary for inclusion in the study (Figure 2).

The baseline characteristics of patients included in the study are described in Table 1. The 461 patients analyzed had a median age of 66 years with an interquartile range of 56–79 years, of which 264 were men (57.3%). Regarding comorbidities, 61% of the patients were hypertensive (of which 23% were taking ACEI), 39.7% had dyslipidemia, 37.3% were diabetic (of which 9.8% were taking DPP-4 inhibitors), 22.2% were obese, and 18% of patients had chronic respiratory disease (6.3% had OSAHS, 6.1% had COPD, and 5.6% were asthmatic). Additionally, 44% of patients had at least mild chronic renal failure and 18.5% of the patients had a glomerular filtration rate lower than 30 mL/min/1.73 m^2^ based on CKD-EPI-09. Almost a third of the patients were taking at least one antiaggregant.

On admission, median SpO_2_/FiO_2_ was 433 with an interquartile range of 239–452 and a median radiological severity of at least moderate. The CALL score value reflected that the disease was at least moderate in most cases.

Regarding the treatment received against COVID-19, the most frequently administered steroid was dexamethasone (6 mg) according to the RECOVERY [35] guideline, which was administered in up to half of the cases. Regarding other treatments, tocilizumab, anakinra, and remdesivir were all used in around 5% of cases. Additionally, 22% of patients received ruxolitinib. Prophylactic doses of LMWH were administered in 80% of cases, intermediate doses in 5.9% of cases, and therapeutic doses in 14.1% of cases (Table 1).

### 3.1. Primary Outcomes

A total of 103 patients died during hospitalization (22.3%). Of the 369 patients who received prophylactic doses, 79 (21.4%) died. Of the 27 individuals who received an intermediate dose, 3 (11.1%) died. Of the 65 cases that received therapeutic doses, 21 died (32.3%).

In the binary logistic regression analysis regarding outcome (discharge or exitus), age, arterial hypertension, diabetes mellitus, dyslipidemia, obesity, OSAHS, COPD, and impaired renal function were observed as risk factors for mortality. Radiological severity at admission calculated with the RALE score was also correlated with an increase in mortality (RR 2.078; 95% CI, 1.242–3.477; *p* = 0.005). This was not the case with CALL score (*p* = 0.22). Regarding the thrombotic risk scales, only the Caprini scale predicted higher mortality (RR 2.754; 95% CI, 1.011–7.501; *p* = 0.048). Regarding laboratory values, nadir prothrombin time (%) was related to lower risk of death (RR 0.939; 95% CI, 0.894–0.985; *p* = 0.01).

There were no significant differences between the different corticotherapy regimens used. The use of anakinra, ruxolitinib, or remdesivir did not show significant differences; nevertheless, the use of tocilizumab was associated with lower mortality (RR 0.056; 95% CI, 0.006–0.545; *p* = 0.013). The use of enoxaparin at therapeutic doses was associated with lower mortality compared to the use of prophylactic and intermediate doses (RR 0.173; 95% CI, 0.038–0.8; *p* = 0.025) (Table 2).

### 3.2. Secondary Outcomes

In total, 33 of the 369 patients who received prophylactic doses (8.9%) required admission to the ICU. Regarding other doses, 7 of the 27 individuals with an intermediate dose (25.9%) and 12 of the 65 cases in the therapeutic dose group (18.5%) required admission to the ICU. Invasive mechanical ventilation was implemented in 23 of the 369 patients administered the prophylactic dose (6.2%), 6 of the 27 administered the intermediate dose (22.2%), and 10 of the 65 administered the therapeutic dose (15.4%). High-flow nasal cannula oxygen was required in 49 patients of the 369 administered the prophylactic dose (13.3%), 4 of the 27 administered the intermediate dose (14.8%), and 18 of the 65 administered the therapeutic dose (27.7%). Adjustment of enoxaparin dose was not associated with fewer admissions to the ICU (*p* = 0.383), neither with the need for invasive mechanical ventilation (*p* = 0.99) nor the use of high-flow nasal cannula oxygen (*p* = 0.723).

Regarding bleeding, in the 369 patients from the prophylaxis group there were 16 minor bleeding episodes (4.3%) and no major bleeding episodes. In the case of intermediate doses, there was only one minor bleeding episode among all 27 patients (3.7%) and no major bleeding episodes. Lastly, in the 65 patients administered therapeutic doses, there were five minor bleeding episodes (7.6%) and four major bleeding episodes (6.1%), of which two led to the death of the patient. In the multinomial logistic regression analysis on bleeding, the IMPROVE bleeding risk score correlated with a higher risk of minor bleeding (RR 1.263; 95% CI, 1.105–1.573; *p* = 0.037), though it was not possible to demonstrate a correlation in the case of major bleeding. No significant differences concerning bleeding could be demonstrated between the three heparin dose groups (Table 3).

During admission, only two cases of DVT could be diagnosed (one in the prophylactic dose group and one in the intermediate dose group), while there were five cases of PE (three in the prophylactic dose group and two in the therapeutic dose group). It should be noted that, among the patients with D-dimer levels greater than 3000 ng/mL, a total of 31 thoracic angioCT scans were performed, of which 26 did not show PE. The thrombotic risk scales did not correlate with thromboembolic events.

## 4. Discussion

Our study aimed to investigate the effect of adjusting the dosage of LMWH on the prognosis of COVID-19 patients. Our findings indicate that adjusting the dosage in specific circumstances can improve the outcome of the disease. Additionally, our study also found that combining clinical variables and laboratory values, such as the IMPROVE bleeding risk scale, can aid in decision making. Overall, our study provides new insights into the use of LMWH in COVID-19 patients.

First of all, it is important to note that the group of patients included in our study is comparable to those in other notable registries in Spain, such as the SEMI-COVID-19 registry [36]. This is particularly true when considering factors such as age and the presence of additional health conditions. Our study population is primarily composed of older individuals and patients with multiple medical issues.

We believe that this aspect of our study is a significant strength as it mirrors the majority of patients that are typically hospitalized with COVID-19. Many previous registries have demonstrated that age and underlying health conditions such as diabetes, hypertension, chronic obstructive pulmonary disease, and obstructive sleep apnea are linked to higher mortality rates for COVID-19 patients [37,38,39]. This is also the case in our study.

Our findings indicate that patients who receive therapeutic doses tend to be older with more underlying health conditions and have a higher degree of respiratory failure, as well as more severe radiologic and laboratory results. Additionally, these patients also have a higher risk of poor outcomes and thromboembolic or hemorrhagic events. This aligns with the protocol followed in our center. The protocol was specifically designed to ensure that patients with a higher risk of severe illness are treated with therapeutic doses. This also reflects the adequate compliance of the protocol.

When comparing our study population to those in other studies investigating the use of low molecular weight heparin in hospitalized patients with SARS-CoV-2 pneumonia [12,17,19], we notice some differences. The patients in our study are older, with an average age of 66 years compared to the average age of 55–60 years in other studies. The population in the therapeutic dose group is even older. Additionally, our patients have a higher number of comorbidities, such as hypertension and diabetes. In other studies, the prevalence of hypertension ranges from 30 to 50% and the prevalence of diabetes ranges from 15 to 30%, while in our study prevalence exceeds 60% and 35%, respectively. Thus, our study population is more similar to the typical population admitted with COVID-19 in clinical practice.

In addition to the comparison of our study population with others, our study also found a significant difference in the mortality rates of COVID-19 patients receiving different doses of enoxaparin. The results of our analysis indicate that the therapeutic dose of enoxaparin is associated with a statistically significant reduction in mortality risk compared to the prophylactic and intermediate doses. Although the confidence interval for the intermediate dose suggests the possibility of unknown benefits, it is clear that the therapeutic dose is the preferred option due to the significant reduction in risk. It is uncertain whether any additional benefit exists with the intermediate dose. It is important to note that the protocol upon which our study is based was developed during the early stages of the pandemic when there was a lack of available evidence and an unusually high incidence of VTE or suspected VTE, which resulted in many deaths.

Additionally, our study found that laboratory values, specifically prothrombin time, are correlated with mortality. The prolonged prothrombin time in patients with more severe illness may be due to the consumption of coagulation factors. Prothrombin time measures the integrity of the extrinsic and common coagulation pathways, and many viral infections have been associated with the activation and elevation of coagulation markers. The increase in these markers indicates a shift towards a procoagulant state, which leads to the consumption or loss of clotting factors, organ failure, and dysfunction in endothelial cells. This finding is consistent with previous studies [40]. Further research has also found that prolonged prothrombin time is associated with an increased risk of mortality [41]. Therefore, our study supports this link.

Similar to our study, other investigations have also found significant differences in terms of reduced mortality when comparing therapeutic doses of heparin against prophylactic doses [19]. However, some of these studies have primarily relied on the D-dimer threshold, which has been criticized as it does not solely reflect the overall clinical condition of the patient and can be affected by various circumstances.

We believe that other cohort studies may not have reached similar conclusions to ours because we managed to reduce the number of potential confounding factors. This includes demographic factors, laboratory results, and other treatments, and also includes the use of the already mentioned risk scales that further strengthens our results.

Indeed, our study takes into account various baseline characteristics of patients, including laboratory values other than D-dimer levels, radiographic severity, and the use of thrombotic and hemorrhagic risk scales. All these factors combined provide a comprehensive perspective for decision making regarding the appropriate dose of LMWH [17].

It is worth noting that the ease of use of the RALE scale allowed for a rapid assessment of radiographic severity at the time of patient admission, which strongly predicts the severity and potential fatal outcome of the disease. Other studies have also demonstrated a strong correlation between this score and clinical severity [42,43].

Our study found that CALL score alone did not determine an increase in mortality during COVID-19 hospitalization. Despite this, we found the CALL score to be a simple and effective tool for quickly classifying patient severity. There is a discrepancy in the literature regarding its usefulness and future studies should further evaluate its validity [22,44].

One of the factors that likely contributed to a lack of significant results in our study population is that many patients with severe COVID-19 were not suitable candidates for invasive mechanical ventilation or ICU admission.

With regards to the prothrombotic risk scales, it is important to note that statistical significance was only observed in relation to a higher risk of mortality when the Caprini scale was used. Significance was not observed when using the Padua or PRETEMED scales. This may be because the Caprini scale places significant emphasis on variables such as age, body mass index, and other comorbidities [45], which are already known to be associated with mortality.

Our study incorporated the IMPROVE hemorrhagic risk score, which we believe to be a strength of our research. The IMPROVE score allowed us to quickly and easily identify patients who were at a higher risk of bleeding, which enabled us to be more cautious when increasing the dose of LMWH in certain cases. Based on our results, we believe the IMPROVE score to be a valid tool for calculating the risk of bleeding in patients with COVID-19 and argue that it accurately reflects the key factors related to COVID-19-associated coagulopathy [46].

Lastly, in regards to the other treatments that were analyzed to eliminate any confounding biases, we found only positive results with tocilizumab. This finding is consistent with some existing literature on the effectiveness of tocilizumab in patients with COVID-19 [47,48].

In relation to the type of corticosteroid therapy used, there are conflicting findings in the literature. Some clinical trials have found no differences in mortality, while others have found that methylprednisolone is associated with lower mortality, despite an increased risk of hyperglycemia associated with its use [49,50]. As this was not a primary focus of our study, we cannot reach a definitive conclusion about the effects of different corticosteroids, although we personally believe that the use of most corticosteroids results in similar outcomes in patients.

Both remdesivir and anakinra were not widely used in our center during the study period. Remdesivir use was limited by its low availability and restrictive use criteria, and anakinra was only used in critically ill patients with low levels of IL-6. Our study found that the use of ruxolitinib did not appear to have any impact on patient outcomes. There are currently no convincing results in the literature on the effectiveness of ruxolitinib [51]. Overall, we believe that it is not possible to draw any conclusions about the effectiveness of treatments other than tocilizumab due to the low number of indications, inconsistent results, and lack of efficacy for those other treatments.

What are the clinical implications of our results? The American Society of Hematology recommends using a prophylactic dose over an intermediate or therapeutic dose for patients with COVID-19-related critical illnesses or acute illnesses without confirmed or suspected thromboembolic disease [20]. Meanwhile, the National Institute for Health and Care Excellence (NICE) guidelines suggest considering a therapeutic dose of LMWH for adults with COVID-19 who require oxygen and who have a low bleeding risk [52]. As a result, the recommendations in the guidelines for dosing LMWH are not consistent.

We concur with these guidelines that early administration of therapeutic heparin in selected cases can affect underlying pathophysiological mechanisms and reduce macrothrombosis and microthrombosis. We also agree that intermediate doses do not appear to be beneficial in this context. Without obtaining the potential advantages of a higher dose, there may be an increased risk of bleeding, although this could not be established in our study. Furthermore, the concept of an intermediate dose can sometimes lead to confusion [53,54].

Given the positive results of our study regarding the effectiveness of the IMPROVE score in identifying patients at lower risk of bleeding, we suggest that it should be considered as a reliable option in guidelines. By using this score, it will be possible to more accurately identify the group of patients who would benefit most from dose adjustment.

Our study has several strengths, such as the similarity of our study population to that of other known registries in Spain (such as SEMI-COVID-19), especially in terms of age and comorbidities, which thus increases the external validity of our research. Additionally, our study was carried out with a consistent protocol agreed upon by one center, and the study population had a greater reduction in terms of confounding factors such as other treatments applied and demographic, radiological, and analytical factors. Additionally, the risk scales used and the inclusion of the IMPROVE hemorrhagic risk score strengthened the results obtained. However, our study also has several limitations that should be taken into account.

One of the limitations of our study is its retrospective nature, which could introduce information and classification biases and which we have attempted to minimize through the use of multiple scales and variables. The fact that it was conducted at a single center could be seen as a limitation or as a strength, although the same protocol was applied to all patients and was interpreted in agreement with the responsible doctors at that time. Some treatments currently in use, such as monoclonal antibodies, were not included in the study, although other treatments such as corticosteroids, remdesivir, and tocilizumab were present. IL-6 levels were not collected for all patients, which could impact the results obtained for tocilizumab. Due to the limited availability of diagnostic tests during the COVID-19 pandemic, it is difficult to draw definitive conclusions about the incidence of VTE or DVT in patients with active SARS-CoV-2 infection. Specifically, the use of ultrasound for DVT diagnosis was hindered by the lack of portable sonographers and the decision not to transfer patients to the Radiology Unit for ultrasound scans in order to minimize exposure to COVID-19. We also did not have a specific screening protocol for this situation (the two cases of DVT were diagnosed after the patients had received negative PCR test results). We suggest that, with proper protective measures in place, the use of portable bedside ultrasound machines should be increased for diagnosing DVT in patients suspected of having COVID-19. This would enable earlier diagnosis and treatment.

It is important to note that the individuals in our study were not vaccinated against the virus, and we did not evaluate the different variants of the virus. As a result, it might be hard to compare our study’s findings to the clinical situation today. This is because, due to increased vaccination and the emergence of less severe strains of the virus, a higher proportion of cases are currently presenting with mild or no symptoms.

In any case, our study is still particularly relevant for unvaccinated immunocompromised individuals or those without adequate levels of SARS-CoV-2 antibodies. It suggests that, even in vaccinated patients, when a prothrombotic state with high inflammation is present, increasing the dose of LMWH in line with our study’s findings may be beneficial.

It is important to note that our study only involved the use of enoxaparin; therefore, it is uncertain whether the results can be extended to other LMWH treatments.

## 5. Conclusions

In adult patients hospitalized with SARS-CoV-2 pneumonia who present with a prothrombotic phenotype, which includes, in addition to elevated D-dimer levels, a severe proinflammatory state, therapeutic doses of enoxaparin should be considered, especially if the risk of bleeding is low according to the IMPROVE bleeding risk score.

## Figures and Tables

**Figure 1 jcm-12-00928-f001:**
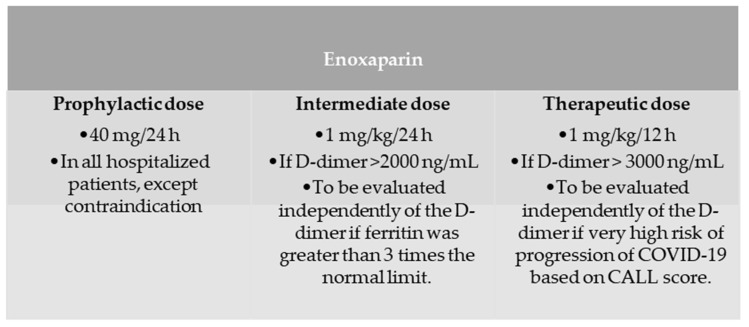
The anticoagulation protocol that was used for hospitalized patients with COVID-19 at Hospital Universitario de Jerez de la Frontera during the period described. CALL Score is based in several clinical and analytical biomarkers [22].

**Figure 2 jcm-12-00928-f002:**
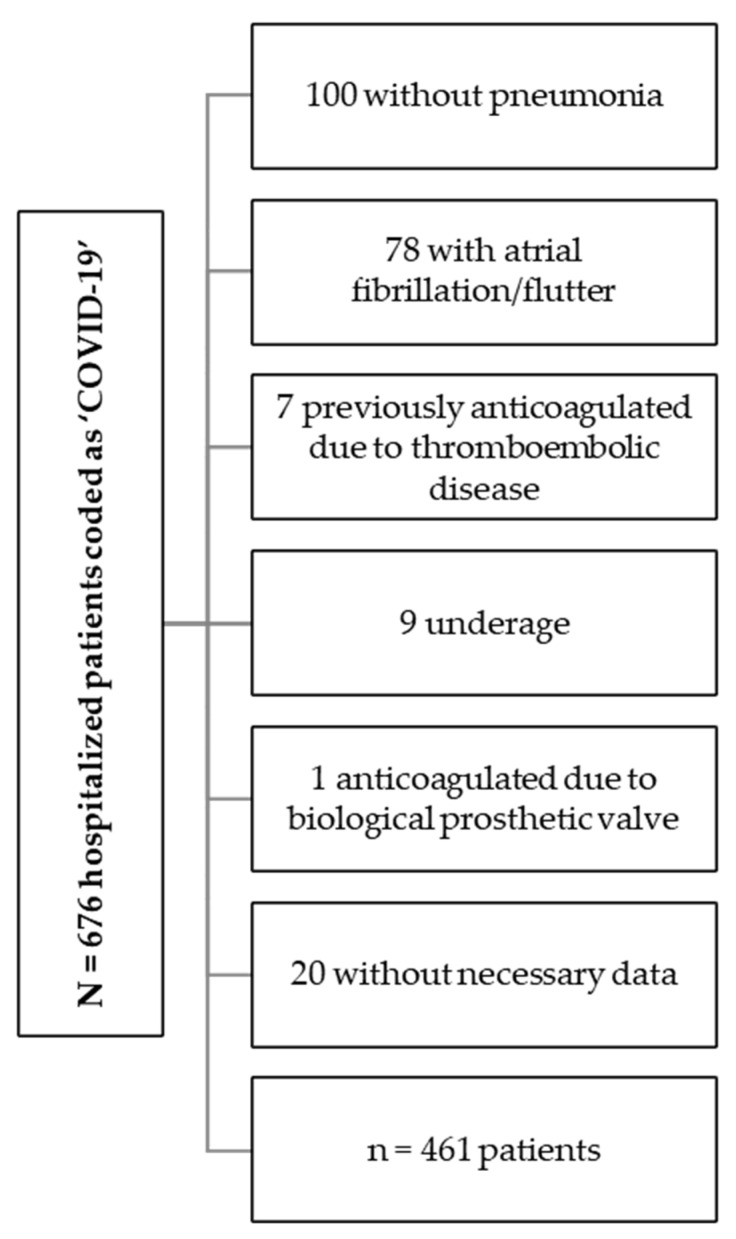
Flowchart of patient inclusion. A total of 215 patients were excluded for the reasons shown in the figure, leaving a total of 461 patients.

**Table 1 jcm-12-00928-t001:** Baseline characteristics of patients included in the study.

n = 461	Prophylaxis (n = 369)	Intermediate (n = 27)	Therapeutic (n = 65)	*p* Value
Age, median (Q1–Q3) (years)	66 (55–78)	64 (54–75)	76 (63–82)	**<0.001**
Men, n (%)	214 (58%)	16 (59.3%)	34 (52.3%)	0.67
**Comorbidities**				
Arterial hypertension, n (%)	224 (60.7%)	13 (48.1%)	44 (67.7%)	0.211
Diabetes mellitus, n (%)	129 (35%)	8 (29.6%)	35 (53.8%)	**0.01**
Dyslipidemia, n (%)	137 (37.1%)	9 (33.3%)	37 (56.9%)	**0.009**
Obesity, n (%)	72 (19.5%)	13 (48.1%)	17 (26.1%)	0.117
Smoking, n (%) Active Previous	27 (7.5%) 84 (23.3%)	2 (8%) 6 (24%)	3 (4.9%) 16 (26.2%)	0.951
Chronic obstructive pulmonary disease, n (%)	22 (6%)	4 (14.8%)	2 (3.1%)	0.098
Sleep apnea/hypopnea syndrome, n (%)	21 (5.7%)	2 (7.4%)	6 (9.2%)	0.539
Asthma, n (%)	19 (5.1%)	1 (3.7%)	6 (9.2%)	0.381
Chronic kidney disease, n (%) G1 (n = 257) (55.8%) G2 (n = 119) (25.8%) G3a (n = 39) (8.5%) G3b (n = 27) (5.8%) G4 (n = 12) (2.6%) G5 (n = 7) (1.5%)	219 (59.3%) 85 (23%) 30 (8.1%) 20 (5.4%) 8 (2.2%) 7 (1.9%)	17 (63%) 7 (25.9%) 2 (7.4%) 0 1 (3.7%) 0	21 (32.3%) 27 (41.5%) 7 (10.8%) 7 (10.8%) 3 (4.6%) 0	**0.011**
**Respiratory severity**				
SpO_2_/FiO_2_, median (Q1–Q3)	375.7 (320–452)	323.8 (198–448)	247.9 (119–419)	**<0.001**
**Radiological severity**				
RALE score, median (Q1–Q3)	5 (4–6)	6 (4–7)	6 (5–7)	**<0.001**
**Scales**				
CALL score, median (Q1–Q3)	10 (8–12)	10 (8–12)	11 (10–12)	**<0.001**
PRETEMED, median (Q1–Q3)	5.26 (3–7)	6.19 (4–8)	7.35 (6–9)	**<0.001**
Padua, median (Q1–Q3)	3.3 (1–5)	4.07 (2–6)	5.28 (4–7)	**<0.001**
Caprini, median (Q1–Q3)	4.86 (3–6)	5.63 (4–7)	5.91 (5–7)	**<0.001**
IMPROVE bleeding risk, median (Q1–Q3)	2.74 (1–3.5)	3.61 (2–5.5)	3.73 (2–5.5)	**<0.001**
**Laboratory test**				
Peak D-dimer, median (Q1–Q3) (ng/mL)	4452.9 (788–2109)	3308.8 (931–5200)	17,938.8 (4144.5–23,338.5)	**<0.001**
Peak activated thromboplastin time, median (Q1–Q3) (ratio)	1.34 (0.94–1.1)	1.19 (0.97–1.15)	1.76 (0.98–1.17)	**<0.001**
Nadir prothrombin time, median (Q1–Q3) (%)	77.8 (70–86)	70.7 (61–79)	73.1 (64.5–81)	**<0.001**
Nadir lymphocytes, median (Q1–Q3) (cels/mm^3^)	959.8 (500–1140)	760 (450–1020)	655.3 (370–795)	**<0.001**
Peak lactate dehydrogenase, median (Q1–Q3) (U/L)	491.6 (284–485)	430 (312–525)	504.3 (345–651)	**<0.001**
Peak C-reactive protein, median (Q1–Q3) (mg/L)	154.8 (74.2–201.7)	158.9 (78–208.3)	203.8 (104.6–250.4)	**<0.001**
Peak procalcitonin, median (Q1–Q3) (ng/mL)	1.34 (0.06–0.31)	0.58 (0.05–0.16)	6.86 (0.1–0.69)	**0.001**
Peak ferritin, median (Q1–Q3) (ng/mL)	1456.7 (361.4–1498.4)	2835.3 (682.8–2687.5)	1582.8 (415.6–1915.9)	**<0.001**
**COVID-19 treatment**				
Steroids, n (%) Dexamethasone 6 mg Dexamethasone 24 mg Methylprednisolone 250 mg	190 (51.5%) 98 (26.6%) 81 (22%)	14 (51.9%) 11 (40.7%) 2 (7.4%)	26 (40%) 25 (38.5%) 14 (21.5%)	0.089
Tocilizumab, n (%)	20 (5.4%)	1 (3.7%)	5 (7.7%)	0.691
Anakinra, n (%)	12 (3.3%)	3 (11.1%)	4 (4.6%)	0.12
Ruxolitinib, n (%)	81 (22%)	10 (37%)	11 (16.9%)	0.105
Remdesivir, n (%)	18 (4.9%)	2 (7.4%)	3 (4.6%)	0.834

SpO_2_/FiO_2_ = relationship between peripheral oxygen saturation and fractional inspired oxygen.

**Table 2 jcm-12-00928-t002:** Analysis of in-hospital mortality according to LMWH dosage and possible confounding variables.

	Relative Risk	95% Confidence Interval	*p* Value
Age	1.077	1.028–1.128	**0.002**
Sex (men)	2.164	0.566–8.278	0.259
**Comorbidities**			
Arterial hypertension	5.304	1.421–19.798	**0.013**
Diabetes mellitus	4.792	1.360–16.877	**0.015**
Dyslipidemia	5.989	1.578–22.734	**0.009**
Obesity	4.320	1.204–15.505	**0.025**
Smoking			
Active	2.248	0.464–10.895	0.315
Previous	9.619	0.673–137.438	0.095
Asthma	1.620	0.051–51.053	0.784
Sleep apnea/hypopnea syndrome	5.123	1.072–24.495	**0.041**
Chronic obstructive pulmonary disease	31.623	2.979–335.625	**0.004**
Chronic kidney disease			0.157
G2	2.937	0.666–12.942	0.155
G3a	15.736	2.103–117.761	**0.007**
G3b	4.914	0.463–52.103	0.186
G4	9.036	0.583–140.001	0.115
G5	9.642	0.187–496.062	0.260
**Respiratory severity**			
SpO_2_/FiO_2_	0.995	0.990–1	0.32
**Radiological severity**			
RALE score	2.078	1.242–3.477	**0.005**
**Scales**			
CALL score	1.338	0.840–2.131	0.22
PRETEMED	0.764	0.361–1.618	0.482
Padua	1.218	0.633–2.343	0.555
Caprini	2.754	1.011–7.501	**0.048**
IMPROVE bleeding risk	0.976	0.662–1.438	0.901
**Laboratory test**			
Peak D-dimer(ng/mL)	1	1	0.06
Peak activated thromboplastin time(ratio)	0.976	0.814–1.171	0.797
Nadir prothrombin time (%)	0.939	0.894-.985	**0.01**
Nadir lymphocytes (cels/mm^3^)	1	0.999–1	0.177
Peak lactate dehydrogenase (U/L)	1.002	1–1.005	0.081
Peak C-reactive protein (mg/L)	1.004	0.999–1	0.117
Peak procalcitonin (ng/mL)	1.047	0.978–1.120	0.187
Peak ferritin (ng/mL)	1	1–1.001	0.058
**COVID-19 treatment**			
Steroids (dexamethasone 6 mg) Dexamethasone 24 mg Methylprednisolone 250 mg	1.823 0.432	0.522–6.36 0.071–2.635	0.346 0.363
Low molecular weight heparin (prophylactic) Intermediate Therapeutic	0.108 0.173	0.011–1.033 0.038–0.8	0.53 **0.025**
Tocilizumab	0.056	0.006–0.545	**0.013**
Anakinra	6.273	0.552–71.301	0.139
Ruxolitinib	0.971	0.281–3.351	0.963
Remdesivir	6.804	0.671–68.995	0.105

SpO_2_/FiO_2_ = relationship between peripheral oxygen saturation and fractional inspired oxygen. The unit used for the variable “age” is 1-year increments. The unit used for the variable nadir prothrombin time is expressed in 1% increments. The unit used for peak activated thromboplastin time is expressed in 0.01 increments.

**Table 3 jcm-12-00928-t003:** Analysis of the different scales in relation to bleeding events.

Bleeding		n of Events	Relative Risk	95% Confidence Interval	*p* Value
**Minor**					
	PRETEMED		1.05	0.787–1.401	0.741
	Padua		1.029	0.738–1.434	0.868
	Caprini		1.079	0.689–1.687	0.741
	IMPROVE bleeding		1.263	1.105–1.573	**0.037**
	Low molecular weight heparin				
	Prophylactic	16	0.702	0.235–2.093	0.525
	Intermediate	1	0.422	0.044–4.022	0.453
	Therapeutic	5	0.652	0.026–3.652	0.657
**Major**					
	PRETEMED		1.046	0.487–2.246	0.909
	Padua		0.865	0.379–1.974	0.730
	Caprini		1.259	0.401–3.955	0.693
	IMPROVE bleeding		1.511	0.837–2.728	0.171
	Low molecular weight heparin				
	Prophylactic	0	*	*	*
	Intermediate	0	*	*	*
	Therapeutic	4	*	*	0.995

* Not enough data for the analysis. Stratification of bleeding according to the ISTH definition [34]. (Major bleeding: fatal bleeding and/or two symptomatic bleeding complications in a critical area or organ, such as intracranial, intraspinal, intraocular, retroperitoneal, intra-articular, pericardial, or intramuscular with compartment syndrome, and/or bleeding causing a fall in hemoglobin level of 2 g/dL or more or leading to transfusion of two or more units of whole blood or red cells. Minor bleeding: nonfatal bleeding in a non-critical area or organ causing a fall in hemoglobin level of less than 2 g/dL without transfusion or only one unit of blood or red cells).

## Data Availability

Data supporting reported results can be shared under request at juanmorainternista@gmail.com.

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
