# Peer review of "Enoxaparin Posology According to Prothrombotic Status and Bleeding Risk in Hospitalized Patients with SARS-CoV-2 Pneumonia"

_jcm, 2023, doi:10.3390/jcm12030928_

Round 1

Reviewer 1 Report

The paper's focus is to determine the optimal dosage of the anticoagulant enoxaparin for the treatment of hospitalized covid-19 patients with COVID pneumonia.  The data used in the paper is based on patients treated in the interval between the beginning of October 2020 and the end of January 2021 in Spain. An advantage of this time period is that the data are not confounded by vaccination status or by prior immunity status, since at that time naturally immunity remained protective enough to prevent repeating infections.

The authors have shown that using the therapeutic dose of enoxaparin results in statistically significant mortality rate reduction, and should be preferred over the intermediate or prophylactic dose. The benefits of using an increased dosage of enoxaparin must be balanced against the increased likelihood of minor or major bleeding. The authors have shown that the IMPROVE bleeding score can be used to identify patients that are at high risk of a minor bleeding episode, but did not have sufficient statistical power to extend this result to major bleeding episodes.

The results of the paper continue to be relevant today, as patients are still being hospitalized due to covid-19, and it's relevance could increase, if future variants prove to be more lethal, although we do not wish that to be the case.

I have a few comments for minor revision

Lines 7-11: The country name "Spain" should be included in the authors affiliations.

Line 147: The dates should be rewritten as "October 1, 2020" and "January 31, 2021" to avoid ambiguity between the American month/day/year format vs the European day/month/year.

Figure 1: This is the first mention of the CALL score in the paper. The authors should add a citation that describes the details of the score.

Line 175: Similarly, the RALE score also needs a citation.

Line 185: The ISTH scale needs a citation as well.

Table 1: The authors should state which statistical tests were used to calculate the p value. This could be done in the table caption. Another approach would be to reference the tables explicitly in section 2.4, and explain it there. Likewise, Table 2 and Table 3 should also be explicitly referenced in section 2.4.

Table 2: In order to interpret the relative risks listed, the authors should clarify whether the unit used for the variable "age" is 1 year increments or whether alternatively they are using sequential age brackets. The units for some of the continuous variables under the heading " Laboratory test" may also need similar clarification (in particular: nadir prothrombin time (is it 1%?); peak activated thromboplastin).

Line 402: Because the confidence interval for intermediate enoxaparin dose is 0.011-1.033, it is more likely than not that there could be some unknown benefit with the intermediate dose as well, so it is not entirely accurate to say that they appear to not be beneficial. The absence of benefit has not been shown. What has been shown is that there is a clear and convincing statistically significant mortality rate risk reduction benefit associated with using the therapeutic dose, so between the two choices the therapeutic dose should be preferred, since we are uncertain whether there is any benefit with the intermediate dose.

Author Response

First of all, we want to thank you for the comments and suggestions made on the text. We certainly believe that the structure of the article is now more appropriate and its content has also increased in quality thanks to your appreciation.

Lines 7-11: added the country.

Line 147: Fixed all three times dates are mentioned in the study.

Figure 1: added corresponding citation.

Line 175: added corresponding citation.

Line 185: added corresponding citation.

Table 1: We have described these aspects in more detail in the text, in the statistical analysis section.

Table 2: we have clearly reflected the required units to facilitate their interpretation. It is just as you supposed.

Line 402: Yes, we think we were too strong with the statement as worded, we think there is now more room for thought regarding intermediate doses given the results.

We are sending you attached the text with the changes (your suggestions are those marked in red. Those marked in yellow are the changes suggested by the other reviewer).

Again, thank you for your time and dedication.

Sincerely,

Juan Mora Delgado

Reviewer 2 Report

I have reviewed the manuscript by Mora-Delgado et al. and there are some concerns deeming discussion. 

1)     Introduction: in the introduction, there are several subheadings which should be deleted or moved to materials and methods. Introduction is usually written presenting the study’s background, rationale, hypothesis and aims. Please modify the structure.

2)     Introduction: “Heparin may have beneficial mechanisms of action in COVID-19 patients, such as direct 59 antiviral effectiveness against SARS-CoV-2.” This sentence must be supported by a reference.

3)     Introduction: “Several studies have compared the effects of therapeutic and prophylactic or intermedi-88 ate-dose anticoagulants. Some reported that low-dose anticoagulants do not significantly 89 improve clinical outcomes compared to intermediate or therapeutic dose anticoagulants, 90 such as the use of invasive or non-invasive mechanical ventilation, ICU admission and death [10–13]. On the other hand, other studies reported that therapeutic doses of LMWH 92 could reduce thromboembolic events and deaths compared to prophylactic treatment or 93 with intermediate doses of heparin in high-risk COVID-19 patients with high D-dimer 94 levels [14–17].” I agree with the authors about this sentence. To date, guidelines recommend against the use of therapeutic dose of anticoagulants (this should be mentioned). In addition, recently some new anticoagulant drug has been also proposed and this should be stated (see PMID: 36294312). Please mention these issues in the manuscript.

4)     Intro: “1.2. Thrombotic and hemorrhagic risk scales” These paragraphs should be moved to another manuscript section.

5)     Introduction: the study aim is not clearly defined. Please provide it.

6)     Materials and methods: the thrombotic and hemorrhagic scales must be defined and described in this section (see above)

7)     Please define minor and major bleedings, as well as all measured outcomes.

8)     Only 2 patients were diagnosed with DVT (that is very low!). Please define how DVT was diagnosed and if a screening ultrasonographic (?) protocol was performed in all patients.

9)     Discussion should start with a short sentence highlighting the novelty of findings based on study aims.

Author Response

First of all, we want to thank you for the comments and suggestions made on the text. We certainly believe that the structure of the article is now more appropriate and its content has also increased in quality thanks to your appreciation.

1) We have modified part of the introduction placing more emphasis on the objectives of the study and our hypothesis. We have modified the structure of the text by moving the scales section to material and methods, thus managing to remove the subsections that unnecessarily segmented part of the text.

2) We have reflected the corresponding reference.

3) We have described the general trend of clinical practice guidelines both in the introduction and in the discussion. We have reflected studies on other anticoagulants as suggested to us.

4) As we have already commented, we have modified the place of this section.

5) We have provided a paragraph describing the hypothesis and objectives of the study.

6) Made change.

7) We have cited the document that reflects the degree of bleeding events. To make Table 3 more self-explanatory, we have described the description of both types of events at the bottom of it.

8) We have placed greater emphasis on the reason that has led us to obtain poor results in this aspect, as well as the limitations and areas for improvement in clinical practice.

9) We have highlighted in the opening paragraph what we consider the greatest strength of our research: the fact of being able to integrate clinical and analytical variables and the validation of a simple-to-apply bleeding risk scale to facilitate decision-making on the dose of LMWH .

We are sending you attached the text with the changes (your suggestions are those marked in yellow. Those marked in red are the changes suggested by the other reviewer).

Again, thank you for your time and dedication.

Sincerely,

Juan Mora Delgado

Round 2

Reviewer 2 Report

No further comments